# Review on Carbon Nanomaterials-Based Nano-Mass and Nano-Force Sensors by Theoretical Analysis of Vibration Behavior

**DOI:** 10.3390/s21051907

**Published:** 2021-03-09

**Authors:** Jin-Xing Shi, Xiao-Wen Lei, Toshiaki Natsuki

**Affiliations:** 1Department of Production Systems Engineering and Sciences, Komatsu University, Nu 1-3 Shicyomachi, Komatsu, Ishikawa 923-8511, Japan; jinxing.shi@komatsu-u.ac.jp; 2Department of Mechanical Engineering, University of Fukui, 3-9-1 Bunkyo, Fukui 910-8507, Japan; lei@u-fukui.ac.jp; 3Faculty of Textile Science and Technology, Shinshu University, 3-15-1 Tokida, Ueda-shi 386-8567, Japan; 4Institute of Carbon Science and Technology, Shinshu University, 4-17-1 Wakasato, Nagano 380-8553, Japan

**Keywords:** carbon nanotubes, carbyne, graphene sheets, nano-force sensor, nano-mass sensor, theoretical analysis, vibration

## Abstract

Carbon nanomaterials, such as carbon nanotubes (CNTs), graphene sheets (GSs), and carbyne, are an important new class of technological materials, and have been proposed as nano-mechanical sensors because of their extremely superior mechanical, thermal, and electrical performance. The present work reviews the recent studies of carbon nanomaterials-based nano-force and nano-mass sensors using mechanical analysis of vibration behavior. The mechanism of the two kinds of frequency-based nano sensors is firstly introduced with mathematical models and expressions. Afterward, the modeling perspective of carbon nanomaterials using continuum mechanical approaches as well as the determination of their material properties matching with their continuum models are concluded. Moreover, we summarize the representative works of CNTs/GSs/carbyne-based nano-mass and nano-force sensors and overview the technology for future challenges. It is hoped that the present review can provide an insight into the application of carbon nanomaterials-based nano-mechanical sensors. Showing remarkable results, carbon nanomaterials-based nano-mass and nano-force sensors perform with a much higher sensitivity than using other traditional materials as resonators, such as silicon and ZnO. Thus, more intensive investigations of carbon nanomaterials-based nano sensors are preferred and expected.

## 1. Introduction

During the last several decades, since the fast development of observation instruments for nanotechnology such as scanning electron microscopy (SEM), transmission electron microscopy (TEM), scanning tunneling microscopy (STM), and atomic force microscopy (AFM), a variety of carbon nanomaterials, e.g., fullerene [1], carbon nanotubes (CNTs) [2,3,4], graphene sheets (GSs) [5], and carbyne [6] were discovered (or predicted) and investigated by scholars (as shown in Figure 1). For instance, according to the invention of STM, which won the Nobel Prize in Physics in 1986, Kroto et al. [1] first produced and observed a soccer ball-like C60 fullerene and were awarded the 1996 Nobel Prize in Chemistry, and Iijima [3] synthesized and measured double-walled, five-walled, and seven-walled CNTs with diameters of 5.5, 6.7 and 6.5 nm, respectively. CNTs with one wall are often called single-walled CNTs (SWCNTs), with two walls are named as double-walled CNTs (DWCNTs), and with more than two walls are named as multi-walled CNTs (MWCNTs). Using SEM and AFM, Novoselov et al. [5] first produced and observed single-layered GSs. Similar to the naming of CNTs, GSs with one layer are often called single-layered GSs (SLGSs), with two layers are named as double-layered GSs (DLGSs), and with more than two layers are named as multi-layered GSs (MLGSs). The present work focuses on the carbon nanomaterials of CNTs and GSs, which are usually adopted as components of nanoelectromechanical systems (NEMS) because of their outstanding material properties.

Regarding the material properties of carbon nanomaterials, the representative one-dimensional (1D) CNTs and two-dimensional (2D) GSs exhibit extremely superior mechanical, thermal, electrical, and optical performance almost on the same level (e.g., [7,8,9,10,11,12,13]). For example, Shokrieh and Rafiee [7] concluded mechanical properties of CNTs determined from both theoretical and experiments methods, and indicated that Young’s modulus of CNTs could reach to the TPa range. Kumar et al. [8] made a review work of material properties of GSs, where they summarized that GSs own Young’s modulus of 1 TPa, thermal conductivity of 1500~5000 W m^−1^K^−1^, electrical conductivity of 10^4^ S/cm, and optical transmittance of 97.7%. Because of their exceptionally high electronic conductivities, the application of CNTs and GSs on transistors [9], nanoelectronics [10], and supercapacitor [11] were also reviewed by scholars. In addition, more detail of their thermal and optical properties could also refer to some of previous review works [12,13]. According to their outstanding material properties introduced above, CNTs and GSs have been proposed and applied as sensing elements in biosensors [14,15,16], strain sensors [17,18,19], and gas sensors [20,21,22]. Besides these three kinds of sensors, carbon nanomaterials are also expected to contribute to the fields of nano-mass and nano-force sensors, which are considered in the present work.

When a mass is too tiny to be detected by a normal measure method, mass sensors using mechanical resonators belonging to micro-electromechanical systems (MEMS), even NEMS, need to be developed. It is well known that mechanical resonators can be used as inertial balances to detect tiny mass by measuring oscillation frequency shifts [23]. Abadal et al. [24] proposed a simple electromechanical model using polysilicon as a cantilevered resonator, which had the sensitivity to detect attogram scales (10^−18^ g) of mass. By means of a complementary metal-oxide-semiconductor (CMOS) circuitry, this model could calculate dynamic quantities of the current flowing through the resonator at the resonance frequency as well as static magnitudes of the collapse voltage and deflection of the resonator, so that the unknown mass could be detected from the feedback of the electrical specifications of the CMOS circuitry as well as the resonance frequency of the resonator. Using the similar approach, silicon-based mass sensors have been investigated and developed further [25,26,27,28,29,30,31,32,33,34,35,36,37]. However, silicon-based mass sensors have their limitations of mass detection due to their relatively lower material properties (e.g., Young’s modulus of 180 GPa [24]) and larger cross-section (e.g., thickness of 1 µm [24]) compared to carbon nanomaterials. Poncharal et al. [38] firstly produced a nano-mass sensor using a cantilevered MWCNT as the resonator and observed the electrically induced dynamic deflections of the resonator attached by a carbon particle. From calculating the resonance frequency as revealed by the deflected contours, they measured the mass of the attached carbon particle to be 22 ± 6 fg (1 fg = 10^−15^ g). After that, plenty of experimental and theoretical investigations of CNTs/GSs-based nano-mass sensors were carried out (e.g., [39,40,41,42,43,44,45,46,47,48]), which owned much higher sensitivity (>10^−21^ g) than silicon-based mass sensors.

Regarding force sensors, for manipulating nano particles, biomolecular or cells, development of high sensitivity force sensors with mechanical types [49,50,51,52,53,54,55,56], electrical types [57,58,59,60,61,62,63,64], and optical types [65,66,67,68], it has been a great challenge for advanced micro/nano-assembly and bio-engineering. Willemsen et al. [49] summarized the works of the detection of biomolecular interaction forces using AFM with silicon nitride probes by that time, in which the AFM probes were considered as force sensors that could detect the interaction forces between individual molecules in nN (10^−9^ N) range by mechanical evaluations, such as strain change or frequency shift. They pointed out that though AFM was a versatile and high enough instrument to discern individual molecules, it could only detect force in one direction, and it would be interesting to be able to measure lateral and torsional forces. Wang et al. [57] demonstrated a piezoelectric field effect transistor (i.e., a nano-force sensor) composed of a ZnO nanowire bridging across two electrodes, which could detect a force in nanonewton range acted on the nanowire by evaluating the decrease of conductance. Hong et al. [66] developed a CNTs-based nano-force sensor composed a CNTs-based transistor suspended with dual-trap optical tweezers, which could detect external forces by monitoring the morphology changes of the transistor using three-dimensional (3D) scanning photocurrent microscopy. This developed nano-force sensor had ability to detect mechanical coupling between individual DNA molecules and the transistor in pN (10^−12^ N) range, which was much more sensitive than silicon nitride/ZnO-based nano-force sensors. More detail of the difference among the three types of force sensors can be found in two previous review works [69,70]. The present work mainly discusses the frequency-based nano-force sensors, i.e., using carbon nanomaterials as resonators for detecting tiny forces from the evaluation of the resonant frequency shifts [71].

Whether for nano-mass or nano-force sensors, the determination of the resonance frequency from the vibration analysis is very important work. Theoretical analysis of the vibration behavior of carbon nanomaterials-based resonators can measure or predict the precisions of proposed nano-mass and nano-force sensors. In the mechanical analysis (e.g., vibration analysis) of carbon nanomaterials, the most essential thing is considering their analytical models. Up to now, there are generally two categories of theoretical model approaches for analyzing the mechanical properties of carbon nanomaterials. The first is the atomistic modeling techniques, such as first-principles calculation (e.g., [72,73,74,75]) and molecular dynamics (MD) simulation (e.g., [76,77,78,79]). Though any mechanical analysis of carbon nanomaterials can be simulated by the atomistic modeling techniques, due to the huge computational tasks, practical execution of the atomistic modeling techniques is often limited to atomistic models with a relatively small number of carbon atoms and a relatively short-lived phenomenon. The second is continuum mechanical modeling approaches, such as beam and shell models of carbon nanomaterials (e.g., [80,81,82,83]), where the appropriate evaluation of material properties is very essential.

The present work aims to review and discuss the recent works about carbon nanomaterials (CNTs/GSs/carbyne)-based nano-mass and nano-force sensors based on vibration analysis using continuum mechanical approaches. In Section 2, we introduce the mechanism of frequency-based nano-mass and nano-force sensors, where we point out the importance of determining the material properties of carbon nanomaterials. In Section 3, we introduce and discuss some representative works for the determination of material properties of continuum models of CNTs, GSs, and carbyne, respectively. In Section 4, we conclude the representative works of CNTs/GSs/carbyne-based nano-mass and nano-force sensors, and propose some challenge works in the future. At last, we summarize the remarkable conclusions.

## 2. Mechanism of Frequency-Based Nano-Mass and Nano-Force Sensors

As described in the introduction, the detection mechanism of frequency-based nano-mass and nano-force sensors is generally based on vibration analysis of the sensor systems for determining the resonant frequency shift, which is sensitive to the resonator force or mass. When an additional mass or an extra force is attached on the resonator of a nano sensor system, the resonant frequency of the resonator changes, so the accurate determination of the variation of the resonant frequency (i.e., the resonant frequency shift) can measure the additional mass or the unknown extra force exactly [41,44,69,71].

### 2.1. Nano-Mass Sensor

In natural vibration (also named as free vibration) analysis of a nano-mass sensor system, the typical governing equation of vibrational motion of resonators for determining its fundamental frequency can be given as [71]:(1)$$\left[M]\{\ddot{y}\}+[K]\{y\}=\{0\right\}$$
where [M] and [K] denote the mass and stiffness matrices of the analytical system, {y} and $\left\{\ddot{y}\right\}$ are the displacement and acceleration vectors, respectively.

Here, we show the mechanism of the nano-mass sensor as following: According to Equation (1), if the dimension (e.g., thickness, diameter, and length and width) and density of resonator related to the mass matrix [M] and the materials properties of resonators (e.g., Young’s modulus, shear modulus, and Poisson’s ratio) related to stiffness matrix [K] are known, the fundamental frequency f of the sensor system without attached mass can be calculated at first. Next, a tiny mass with mass matrix [ΔM] is added on the resonator to generate a different mass matrix [M+ΔM] of the total sensor system and a new fundamental frequency $f^{\prime }$ can be determined, so the frequency shift $\Delta f=f^{\prime }-f$ can be determined. Repeating this process by alternating the mass matrix [ΔM] of the tiny mass, the correlation curve between the additional mass and the frequency shift of the total sensor system performs one-to-one correspondence, which is used to measure an unknown tiny mass at last. However, when the tiny mass becomes smaller and smaller, the frequency shift shows a tiny change that cannot be recognized clearly, the nano-mass sensor reaches its limitation of mass detection.

### 2.2. Nano-Force Sensor

For governing equation of a nano-force sensor system that detecting unknown external forces, a loading vector {F} is added on the right side of Equation (1), shown as
(2)$$\left[M]\{\ddot{y}\}+[K]\{y\}=\{F\right\}$$

We introduce the mechanism of the nano-force sensor as following, which is similar with that of the nano-mass sensor. At first, the frequency f of the resonator without external force (i.e., {F}={0} shown as Equation (1)) should be theoretically calculated. Then, a given external force related to the loading matrix {F} is added on the resonator to generate a new frequency $f^{\prime }$ from Equation (2) for determining the frequency shift $\Delta f=f^{\prime }-f$ or the relationship between the external force and the frequency. Accordingly, the correlation curve between the external force and the frequency shift (or the frequency) of the total sensor system can be drawn, which can be used to measure an unknown external force at last. Additionally, when the external force becomes smaller and smaller until the frequency shift shows an extremely tiny change that cannot be recognized, the nano-force sensor reaches its measure limitation of force detection.

From Section 2.1 and Section 2.2, we can find that, if we want to use Equations (1) and (2) for vibration analysis of carbon nanomaterials-based nano-mass and nano-force sensors, the dimensions (e.g., thickness, diameter or width, and length) and densities of carbon nanomaterials-based resonators and the materials properties of the resonators (e.g., Young’s modulus, shear modulus, and Poisson’s ratio) should be determined coinciding with the analytical models of nano-mass and nano-force sensors. Thereby, plenty of works have been carried out for determining the material properties of continuum models of carbon nanomaterials.

## 3. Continuum Models of Carbon Nanomaterials

Up to now, different continuum models have been adopted for carbon nanomaterials. According to Equations (1) and (2), material properties (e.g., Young’s modulus, shear modulus, and Poisson’s ratio) as well as the dimensions (e.g., thickness, diameter or width, and length) of different continuum models of carbon nanomaterials should be determined appropriately. Here, we introduce some representative works for determining the material properties and corresponding dimensions of continuum models of CNTs, GSs, and carbyne, respectively.

### 3.1. Carbon Nanotubes and Graphene Sheets

Among carbon nanomaterials, CNTs and GSs have been attracted the most interest of scholars. In general, there are two equivalent continuum models, which are shell and beam models, usually adopted in theoretical analysis of CNTs and GSs. Generally, perfect GSs are 2D materials and CNTs can be considered as tubes rolled from GSs, so both of CNTs and GSs are often analyzed based on classical plate/shell theories (e.g., [84,85,86,87,88]). Additionally, CNTs are 1D materials, so beam theories can be adopted for mechanical analysis of CNTs (e.g., [89,90,91,92]), which is used the most in theoretical evaluation of CNTs-based nano-mass and nano-force sensors. Moreover, graphene nanoribbons (GNRs) are special kinds of GSs owning 1D structures, so mechanical behavior of GNRs can be also analyzed by beam theories (e.g., [93,94,95,96]). Whether using shell theories or beam theories for mechanical analysis of CNTs and GSs, their material properties should be evaluated matching with atomistic modeling techniques.

With respect to CNTs, Yakobson et al. [76] performed MD simulation of SWCNTs subjected to axial compressive forces and estimated Young’s modulus and thickness of shell model of SWCNTs as 5.5 TPa and 0.066 nm, which could be used for theoretical analysis of mechanical behaviors of SWCNTs. However, this work was discussed as the well-known “Yakobson’s paradox” because of the contradicting results of Young’s modulus of SWCNTs compared with other studies (around 1 TPa) [97,98,99].

In 2003, Li and Chou [100] proposed a notable continuum mechanical approach, named as molecular structural mechanics (MSM), for modeling CNTs and GSs with frame-like structures by establishing a linkage between the molecular mechanics (MM) and the structural mechanics. Detail of this approach was shown in Figure 2, where the total steric potential energy U of each C–C chemical bond with a summation of the bond stretching interaction energy $U_{l}$, the bond angle bending energy $U_{\theta }$, and the equivalent torsion energy $U_{\tau }$ were expressed as shown in Equation (3).
(3)$$U=\sum U_{r}+\sum U_{\theta }+\sum U_{\tau }$$
and each potential energy was given as:(4)$$U_{r}=\frac{1}{2}k_{l}{\left(\Delta l\right)}^{2}$$
(5)$$U_{\theta }=\frac{1}{2}k_{\theta }{\left(\Delta \theta \right)}^{2}$$
(6)$$U_{\tau }=\frac{1}{2}k_{\tau }{\left(\Delta \phi \right)}^{2}$$
where $k_{l}$, $k_{\theta }$, and $k_{\tau }$ indicated the bond stretching resistance constant, the bond bending resistance constant, and the bond torsion resistance of a C–C chemical bond, respectively, which were determined based on MM [101]. Δl, Δθ, and Δϕ were the stretching deformation, the bending rotational angle, and the torsion rotational angle, respectively.

On the other hand, each C–C chemical bond could be also assumed as an equivalent continuum beam with the tensile resistance $E_{e}A$, the flexural bending rigidity $E_{e}I$, and the torsion stiffness $G_{e}J$. Accordingly, each potential energy in terms of stretching, bending, and torsion energy of the equivalent continuum beam could also be determined from structural mechanics, shown as:(7)$$U_{r}=\frac{1}{2}\frac{E_{e}A}{l}{\left(\Delta l\right)}^{2}$$
(8)$$U_{\theta }=\frac{1}{2}\frac{E_{e}I}{l}{\left(\Delta \theta \right)}^{2}$$
(9)$$U_{\tau }=\frac{1}{2}\frac{G_{e}J}{l}{\left(\Delta \phi \right)}^{2}$$
where l is the length of the C–C chemical bond.

Using the direct relationship between Equations (4)–(6) and Equations (7)–(9), the material properties and sectional stiffness parameters of the equivalent continuum beam could be calculated and adopted in FEM for mechanical analysis of CNTs and GSs. With this approach, Li and Chou calculated Young’s moduli and shear moduli of GSs and SWCNTs with different diameters, which were 0.85~1.05 TPa and 0.2~0.5 TPa, respectively, as shown in Figure 3. Up to now, the approach of MSM has been widely adopted for calculating the material properties (e.g., Young’s modulus and shear modulus) of CNTs (e.g., [102,103,104,105,106,107,108,109]) and GSs (e.g., [106,107,108,109,110,111,112,113,114,115,116]) for mechanical analysis with continuum shell and beam models.

Along with the Young’s modulus and shear modulus, the thickness of the continuum models of CNTs and GSs was also an essential dimensional parameter in theoretical analysis. 0.34 nm, the interlayer distance between each graphene layer in graphite, was commonly assumed for mechanical analysis of CNTs and GSs (e.g., [117,118,119,120]. However, for the mechanical analysis of CNTs and GSs using continuum mechanics theories, the appropriate thickness should be determined matching with different analytical conditions. For example, in the “Yakobson’s paradox”, the thickness of CNTs was calculated as 0.066 nm (paired with Young’s modulus 5.5 TPa) matching with MD simulation under loading condition of axial compression. Shi et al. [112] calculated Young’s modulus and thickness of GSs as 2.81 TPa and 1.27 Å considering bending and stretching loading conditions simultaneously. However, both works did not consider the density term, so they are difficult to be adopted in vibration analysis of CNTs and GSs. Hence, we should note here that, in vibration analysis of CNTs and GSs, the density term related to the mass matrix usually should be determined simultaneously matching with Young’s modulus, shear modulus, and thickness, which will be an essential work in development of CNTs and GSs-based nano sensors using continuum mechanical approaches in the future.

### 3.2. Carbyne

Carbyne is a chain of carbon atoms linked with double chemical bonds (…C=C=C…) or alternating single and triple chemical bonds (…C–C≡C…). Liu et al. [121] investigated the mechanical behavior of carbyne by means of first-principles calculations, where they modeled the carbyne as an elastic beam and established a link between the molecular model and the continuum beam model of carbyne under tension, bending, and torsion loading conditions. In this landmark study, they calculated the diameter, Young’s modulus, shear modulus, and Poisson’s ratio of the equivalent continuum beam model of carbyne as 0.772 Å, 32.71 TPa, 11.8 TPa and 0.386, respectively (The authors corrected the shear modulus and Poisson’s ratio as 47.2 TPa and –0.65 later [122]). These material properties have been adopted for most of the theoretical studies of carbyne using continuum mechanical approaches (e.g., [123,124,125]).

For the density of carbyne $\rho_{carbyne}$ that can be used in vibration analysis, Shi et al. [123] calculated it as 32.21 g/cm^3^ from the following equation:(10)$$\rho_{carbyne}=\frac{4m_{carbyne}}{\pi D_{carbyne}^{2}L_{carbyne}}$$
where $m_{carbyne}$ is the total mass of a carbyne with 12 carbon atoms, $D_{carbyne}$ is the diameter of the equivalent continuum carbyne beam, $L_{carbyne}$ is the length of the carbyne with 12 carbon atoms.

## 4. Nano-Mass Sensor

With the material properties determined from Section 3, carbon nanomaterials-based nano-mass sensors can be investigated by continuum mechanical approaches using the mechanism shown in Section 2.1. Here, we introduce and summarize representative works of CNTs, GSs, and carbyne-based nano-mass sensors, respectively, by vibration analysis based on their continuum models.

### 4.1. Carbon Nanotubes-Based Nano-Mass Sensor

CNTs-based nano-mass sensors have been studied using variety of theoretical mechanical approaches, such as FEM (e.g., [126,127,128,129,130,131,132]) and continuum beam theories (e.g., [127,129,131,133,134,135,136,137,138,139,140,141,142,143,144]). Though shell theories of CNT were adopted in vibration analysis of CNTs (e.g., [84,85,86,87,88]), they usually treated CNTs with small aspect ratio (length/diameter) and have been seldom used for study of CNTs-based nano-mass sensor to our knowledge. The reason can be considered as that only the first vibrational mode can be evaluated for the application of nano-mass sensor and most of the CNTs-based resonators have large aspect ratio (length/diameter). Thereby, using beam models of CNTs is simple and sufficient to evaluate the efficiency of CNTs-based nano-mass sensors.

Li and Chou [126] studied CNTs-based nano-mass sensors by adopting the approach of MSM for modeling CNTs and GSs that just proposed by themselves [100]. In this study, SWCNTs with length 10 nm and diameters of 0.8, 1.0 and 1.2 nm were proposed as the resonators, and two boundary conditions, cantilevered and bridged, were considered as shown in Figure 4. From their results, as shown in Figure 5, the resonant fundamental frequencies of both cantilevered and bridged CNTs decreased with the increase of attached mass, and when the attached mass was larger than 10^−6^ fg (i.e., 10^−21^ g), a logarithmically linear relationship between the resonant frequency and the attached mass could be found, which means that the proposed CNTs-based nano-mass sensor had ability to measure a tiny mass larger than 10^−21^ g. In addition, the frequency shift increased with the increase of attached mass, and shorter (as shown in Figure 6a) or thicker (as shown in Figure 6b) CNTs resonator owned higher mass sensitivity. However, comparing with Figure 6a,b, the effect of tube length was much bigger than that of tube diameter on the sensitivity.

We summarize some of the representative works with respect to CNTs-based nano-mass sensor adopting FEM [126,127,128,129,130,131,132], Euler–Bernoulli beam theory (EBT) [127,129,131,135,138,140], nonlocal EBT [133,134,136,139,142,143,144], Timoshenko beam theory (TBT) [141], nonlocal TBT [137] for studying CNTs-based nano-mass sensors as shown in Table 1, from which we can see that though the work considering thermal and nonlocal effects could make sensitivity to atom mass of 6.65 × 10^−24^ g and 0.218 × 10^−24^ g under special thermal conditions [143,144], most of the works indicated that the CNTs-based nano-mass sensor could detect tiny mass larger than 10^−21^ g. Moreover, detail of CNTs-based nano-mass sensor can be also found in some previous review works [18,145].

### 4.2. Graphene Sheets-Based Nano-Mass Sensor

The study of GSs-based nano-mass sensors using continuum mechanical approaches is also popular among scholars. In contrast to 1D CNTs, GSs own 2D structures in general. Hence, using the material properties introduced in Section 3.1, continuum plate/shell theories are often adopted by scholars, such as FEM (e.g., [146,147,148]) and elasticity plate theory (EPT) (e.g., [149,150]). Tsiamaki et al. [146] proposed a circular GSs-based nano-mass sensor and simulated its vibration behavior using FEM for calculating the frequency shift. They discussed different boundary conditions of the GSs resonators and compared their results with other works to demonstrate the reasonable accuracy of the results. Their results showed that the proposed nano-mass sensor had sensitivity of 10^−22^ g level. Natsuki et al. [149] presented a frequency-based nano-mass sensor using rectangular DLGSs as resonators, where a continuum EPT was adopted for vibration analysis and sensitivity of the presented nano-mass sensor could also reach 10^−22^ g level. Furthermore, nonlocal EPT that considering nonlocal effects is also popular for the studies of GSs-based nano-mass sensors (e.g., [151,152,153,154,155,156,157]). Shen et al. [151] modeled a simply supported SLGSs-based nano-mass sensor and calculated its frequency shifts using the nonlocal Kirchhoff plate theory. The mass sensitivity of the SLGSs-based nano-mass sensor could reach at least 10^−21^ g. They also pointed out that the frequency shifts became smaller when the nonlocal effect was considered. In addition, as special kinds of GSs that own 1D structures, GNRs-based nano-mass sensors were also investigated using EBT (e.g., [158]) or nonlocal EBT (e.g., [159]).

To show a clear comparison, the results of GSs-based nano-mass sensors are summarized in Table 2. The summarization shows that most works indicated GSs-based nano-mass sensors have ability to detect tiny mass from 10^−24^ g to 10^−22^ g at least, which is more sensitive than CNTs-based nano-mass sensors.

### 4.3. Carbyne-Based Nano-Mass Sensor

Though carbyne really exists or not was a subject of great interest among some scholars several decades ago [160,161,162], carbyne has been proposed with a higher stiffness than CNTs and GSs [121]. It is also expected that resonators made by carbyne can perform higher sensitivity in nano-mass sensors.

Using the material properties determined by Liu et al. [121], Shi et al. [123] predicted the sensitivity of a carbyne-based nano-mass sensor. Considering the difficulty of preparation of long carbyne chain, carbyne resonators with 12 carbon atoms and 17 carbon atoms were investigated in this work, where they adopted the nonlocal TBT for studying the two kinds of carbyne resonators with low aspect ratios. Moreover, they also performed the Rayleigh energy method and MD simulation to confirm the feasibility of the nonlocal TBT, where the results obtained from the three methods coincided each other very well. To obtain the highest sensitivity of the carbyne-based nano-mass sensor, initial stressed carbyne resonators were also studied, where they found that a higher initial stress could obtain higher fundamental frequency of the resonator as well as higher sensitivity of mass detection. According to their results, carbyne-based nano-mass sensor could detect tiny mass reaching to the range of 10^−26^ g.

Just after the above-mentioned work was published, Agwa and Eltaher [124] studied the carbyne-based mass-sensor considering the influence of surface effects (i.e., surface stress and surface elasticity) on the vibration behavior of a carbyne resonator with 12 carbon atoms, where they adopted the TBT in this work. According to their results, the surface stress and surface elasticity had considerable effect on vibration behavior of the carbyne resonator, and the proposed carbyne-based nano-mass sensor had ability to detect a tiny mass below 10^−23^ g.

Hence, we can confirm that carbyne-based nano-mass sensors own the highest sensitivity of detecting tiny mass among the three kinds of carbon nanomaterials.

## 5. Nano-Force Sensor

Similar to the nano-mass sensors, using the mechanism shown in Section 2.2 and the material properties determined from Section 3, carbon nanomaterials-based nano-force sensors can also be investigated by continuum mechanical approaches. However, up to now, the studies of theoretical analysis about carbon nanomaterials-based nano-force sensors are only a few to our knowledge (e.g., [71,92,163]).

To interest more scholars in devoting themselves to the study of carbon nanomaterials-based nano-force sensors using theoretical analysis, we emphasize a previous study of CNTs-based nano-force sensors carried out by Natsuki and Urakami to show its feasibility [92]. This study was performed based on vibration analysis of CNTs using a continuum mechanical approach. In detail, a SWCNT with diameter D = 2 nm and length L = 40 nm as shown in Figure 7 was considered as the probe of an AFM (i.e., the resonator of a nano-force sensor). The CNT probe was partial embedded in epoxy resin with embedded length $L_{1}$ and exposed length $L_{2}$ for detecting an unknown external compressive force N.

Considering the large aspect ratio of the CNT probe, it was schemed as a continuum beam model and the epoxy resin was described as an elastic medium. When the CNT beam generated flexural deflection w, the interaction pressure p between the CNT beam and the surrounding elastic medium was described by springs with constant of $k_{w}$ according to the Whitney–Riley model, shown as:(11)$$p=k_{w}w$$

For vibration analysis, the governing equation was expressed as Equation (12) based on EBT.
(12)$$EI\frac{\partial^{4}w}{\partial x^{4}}+N\frac{\partial^{2}w}{\partial x^{2}}+\rho \frac{\pi D^{2}}{4}\frac{\partial^{2}w}{\partial t^{2}}=p$$
where E and I indicate Young’s modulus and moment of inertia of the CNT beam, respectively. x and t denote the longitudinal coordinate and time, respectively. Hence, the vibration motion of the embedded part and exposed part of the CNT beam were performed as Equations (13) and (14), respectively.
(13)$$EI\frac{\partial^{4}w_{1}}{\partial x^{4}}+N\frac{\partial^{2}w_{1}}{\partial x^{2}}+\rho \frac{\pi D^{2}}{4}\frac{\partial^{2}w_{1}}{\partial t^{2}}=-k_{w}w_{1},\;w_{1}\in [0,L_{1}]$$
(14)$$EI\frac{\partial^{4}w_{2}}{\partial x^{4}}+N\frac{\partial^{2}w_{2}}{\partial x^{2}}+\rho \frac{\pi D^{2}}{4}\frac{\partial^{2}w_{2}}{\partial t^{2}}=0,\;w_{2}\in [L_{1},L]$$
where $w_{j},j=1,2$ are the flexural deflections of the embedded part and the exposed part of the CNT beam.

Then, using a mathematical technique to solve the two differential equations and considering the boundary conditions at $x=0,L_{1},\mathrm{and}L$, a simultaneous equation was obtained as:(15)$$\Omega {\left[N,L_{1},L_{2}\right]}_{8\times 8}\left[\begin{array}{c} A_{1}\\ A_{2}\\ \vdots \\ A_{8} \end{array}\right]=0$$
where $\Omega {\left[N,L_{1},L_{2}\right]}_{8\times 8}$ is a 8 × 8 matrix in terms of the external compressive force N, the embedded length $L_{1}$, and the exposed length $L_{2}$ of the CNT beam. $A_{j},j=1,2,\cdots ,8$ are integration constants that can be determined from the governing equations of vibration motion of the CNT beam by considering boundary conditions. From the non-trivial solution of Equation (15), when the eigenvalue of $\Omega {\left[N,L_{1},L_{2}\right]}_{8\times 8}$ becomes 0, i.e., $\Omega {\left[N,L_{1},L_{2}\right]}_{8\times 8}=0$, the relationship between the external compressive force and the frequency can be obtained.

According to the non-trivial solution of Equation (15), the first three vibrational modes of the partial embedded CNT beam under an external compressive force were obtained and expressed in Figure 8 as a result. Furthermore, the relationship between the external compressive force and the frequency of vibrational modes 1 and 2 were drawn in Figure 9, the results showed that the fundamental frequency of the CNT beam decreased clearly as the external compressive force increased. Hence, by calculating the frequency shift from the obtained relationship curve of mode 1, the unknown compressive force can be detected theoretically. However, for the compressive loading condition, critical force of 2.5 nN appeared in mode 1, which means the proposed CNTs-based nano-force sensor had its upper limitation of 2.5 nN of force detection.

In the future, just like the representative works of theoretical analysis of nano-mass sensors summarized in Section 4, because of their extremely excellent material properties, CNTs/GNRs/carbyne-based nano-force sensors investigated by approaches of FEM, EBT, etc., and GSs-based nano-force sensors investigated with FEM, EPT, etc., are expected for the development of carbon nanomaterials-based nano-force sensors. Additionally, nonlocal elasticity theory is preferred to be considered for deep investigation with theoretical analysis. Moreover, the measurement of external forces acting in different directions is also an important challenge for real application of nano-force sensors, where the vibration behavior of carbon nanomaterials resonators will become more complicated.

## 6. Conclusions

In summary, due to the high-speed development of nanotechnology in the field of nano sensors, the present work reviewed recent studies of frequency-based carbon nano-mass and nano-force sensors using carbon nanomaterials as resonators by continuum mechanical approaches. Three kinds of carbon nanomaterials, CNTs, GSs (including GNRs), and carbyne were considered as resonators of sensors, and the efficiency of each carbon nanomaterial was summarized and discussed. We have listed the highlights of this review work as the following:The mechanism of nano-mass and nano-force sensors based on vibration analysis were introduced theoretically.The methods of modeling CNTs, GSs, and carbyne as continuum structures were reviewed in detail. Especially, we have proposed that, in the vibration analysis of CNTs and GSs, besides Young’s modulus, shear modulus, and thickness, their densities should be determined simultaneously, which will be an essential work for studying CNTs/GSs-based nano sensors in the future.By summarizing the recent studies of carbon nanomaterials-based nano-mass sensors, CNTs, GS, and carbyne-based nano-mass sensors owned the minimum sensitivity of 10^−23^ g, 10^−24^~10^−22^ g, and 10^−26^~10^−23^ g, respectively. Hence, nano-mass sensors using carbyne resonators can provide the highest sensitivity among the three kinds of carbon nanomaterial resonators.Carbon nanomaterials-based nano-force sensors are seldom investigated. However, because of their extremely excellent material properties, CNTs/GSs/carbyne-based nano-force sensors should be studied further by vibration analysis. Moreover, discussion of detecting external forces acting in different directions would also be a deserved work toward the real application of nano-force sensors in the future.At present, the nanobalance technique for measuring the frequency shift of CNTs was demonstrated that could be applied to measure the mass of a tiny particle of light as 22 × 10^−15^ g [38]. Fifty-one gold atoms loaded on CNTs resonators could be experimentally measured using the relationship between the resonance frequency and atom numbers [40]. However, the real-time application of the nano-testing techniques would be a big challenge due to the small size and weight of carbon nanomaterials. New methods and approaches should be well established to reduce the measurement uncertainly and increase testing accuracy.

## Figures and Tables

**Figure 1 sensors-21-01907-f001:**
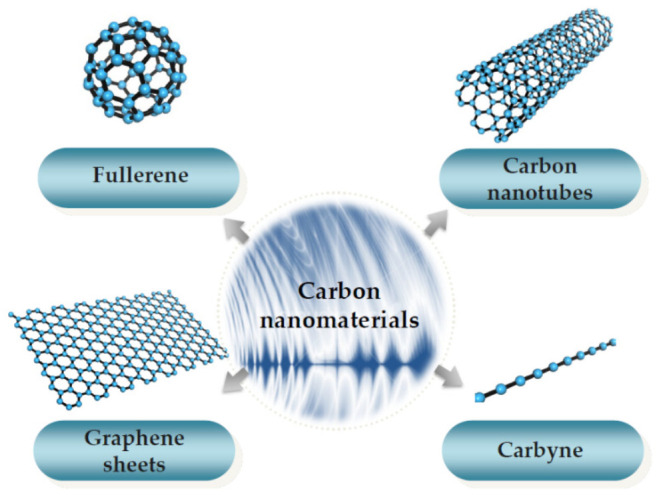
Kinds of carbon nanomaterials.

**Figure 2 sensors-21-01907-f002:**
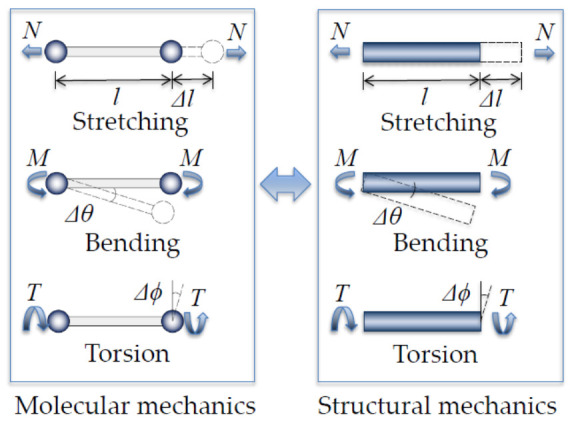
Continuum mechanical approach for modeling C–C chemical bond as an equivalent continuum beam from a linkage between molecular mechanics and structural mechanics.

**Figure 3 sensors-21-01907-f003:**
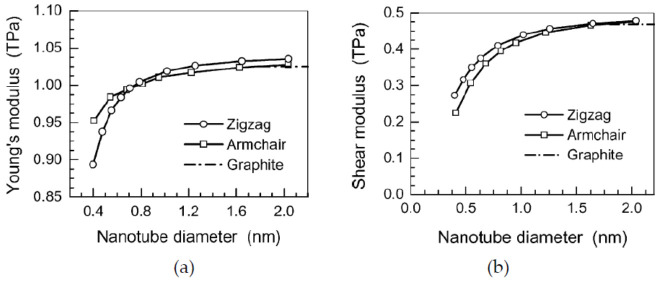
(**a**) Young’s moduli and (**b**) shear moduli of carbon nanotubes versus tube diameter. Adapted with permission for [100], copyright Elsevier, 2003.

**Figure 4 sensors-21-01907-f004:**
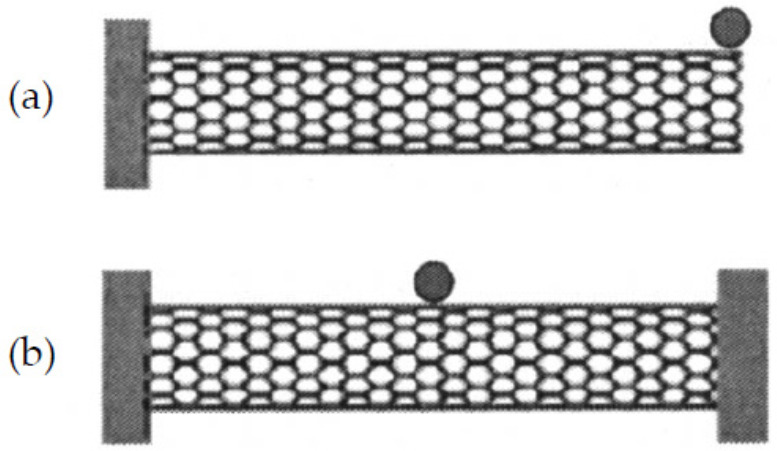
(**a**) Cantilevered and (**b**) simply supported carbon nanotube resonators with an attached mass. Adapted with permission for [126], copyright AIP Publishing, 2004.

**Figure 5 sensors-21-01907-f005:**
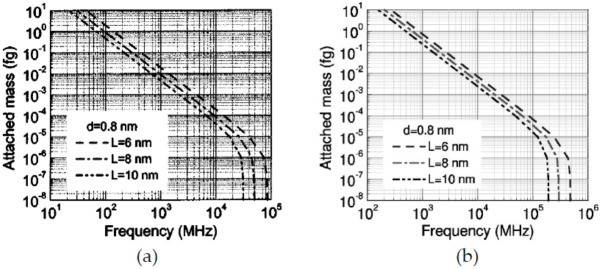
Fundamental frequency of (**a**) cantilevered and (**b**) bridged carbon nanotube resonators with different length L vs. attached mass. Adapted with permission for [126], copyright AIP Publishing, 2004.

**Figure 6 sensors-21-01907-f006:**
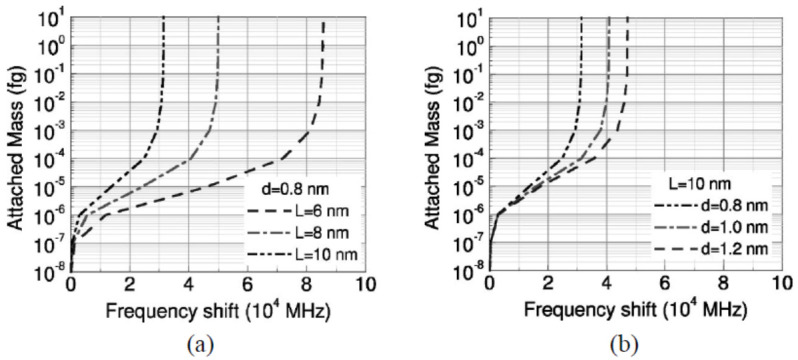
Frequency shift of cantilevered carbon nanotube resonators with (**a**) different lengths L or (**b**) different diameters d vs. attached mass. Adapted with permission for [126], copyright AIP Publishing, 2004.

**Figure 7 sensors-21-01907-f007:**
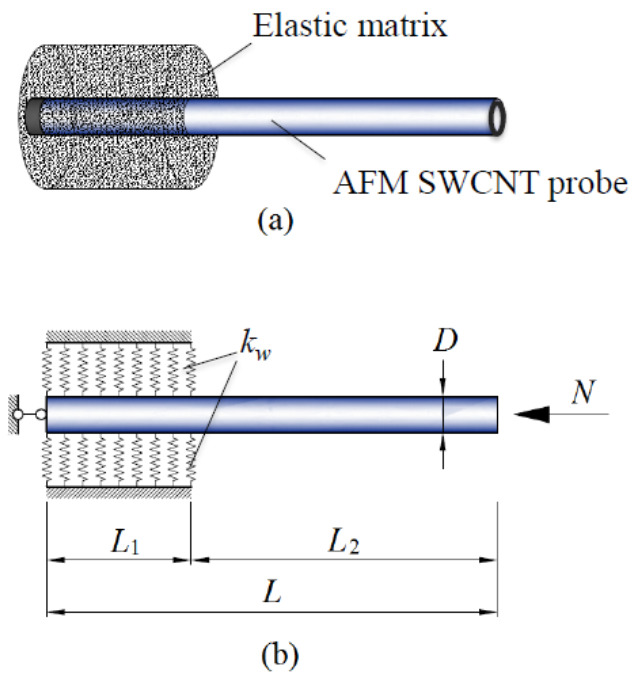
(**a**) A proposed carbon nanotubes-based nano-force sensor, and (**b**) the analytical model of the partial embedded carbon nanotubes resonator [92].

**Figure 8 sensors-21-01907-f008:**
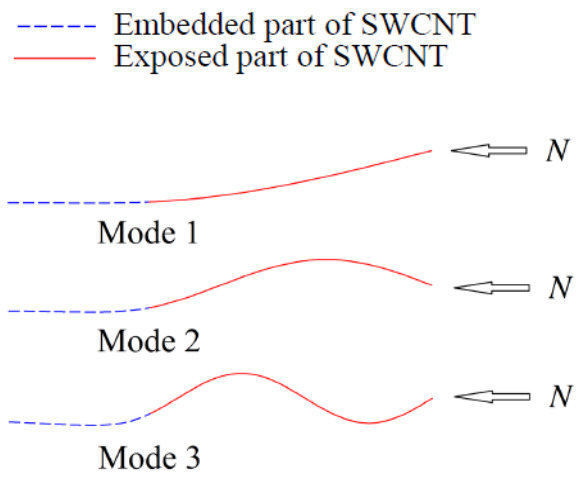
The first three vibrational modes of the partial embedded carbon nanotubes resonator under an external compressive force [92].

**Figure 9 sensors-21-01907-f009:**
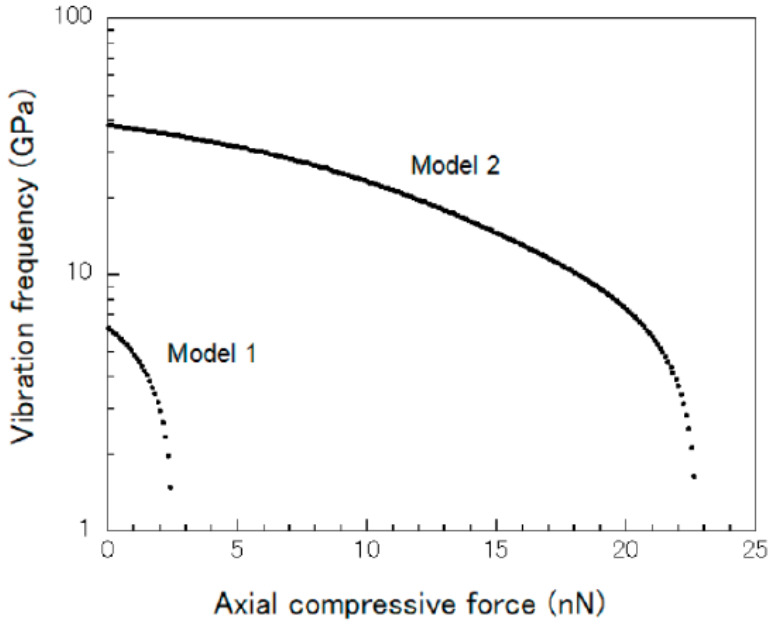
Relationship between the external compressive force and the frequency of the partial embedded carbon nanotubes resonator [92].

**Table 1 sensors-21-01907-t001:** Summarization of studies on carbon nanotubes-based nano-mass sensors.

Author (s)	Information of CNTs	Boundary Condition (s)	Methods	Sensitivity
Li and Chou [123]	SWCNTs with a diameter of 0.8 nm, lengths of 6, 8 and 10 nm	Cantilevered, bridged	MSM, FEM	>10^−21^ g
Wu et al. [127]	SWCNTs with diameters of 24.9, 37.55 and 44.9 nm, lengths of 5.55, 4.65 and 5.75 µm	Cantilevered	EBT, FEM	>10^−21^ g
Li et al. [128]	[6,0]@(6,0) Super CNTs with diameters of 10.81, 7.99 and 5.17 nm, lengths of 29.273, 21.6 and 13.926 nm	Cantilevered, bridged	MSM, FEM	Super CNTs is 6.2~8.87 times of SWCNTs
Chowdhury et al. [129]	SWCNTs with diameter of 1.1m, lengths of 4.1, 5.6 and 8.0 nm	Cantilevered, bridged	EBT, FEM	>10^−21^ g
Georgantzinos and Anifantis [130]	SWCNTs with diameters of 0.54, 0.8 and 1.09 nm, lengths of 6, 8 and 10 nm DWCNTs with inner diameter of 0.41, 1.09 and 1.76 nm, outer diameter of 2.44 nm, length of 17 nm	Cantilevered, bridged	MSM, FEM	SWCNTs is 2 times of DWCNTs
Joshi et al. [131]	SWCNTs with diameter of 0.8 nm, lengths of 6, 8 and 10 nm	Cantilevered, bridged	EBT, FEM	>10^−21^ g
Cho et al. [132]	SWCNTs with diameter of 2.7 nm, length of 55 nm	Cantilevered, bridged	FEM	>2 × 10^−18^ g
Lee et al. [133]	SWCNTs with diameter of 1.1 nm, lengths of 4.1, 5.6 and 8.0 nm	Cantilevered	Nonlocal EBT	>10^−21^ g
Aydogdu and Filiz [134]	SWCNTs with diameter of 1 nm, length of 10 nm	Cantilevered, bridged	Nonlocal EBT	>10^−21^ g
Mehdipour et al. [135]	SWCNTs with diameter of 25.3 nm, length of 5.5 µm	Cantilevered	EBT	>2 × 10^−14^ g
Shen et al. [136]	SWCNT with diameter of 1.05 nm, lengths of 14, 28 and 42 nm DWCNTs with inner diameter of 0.7 nm, outer diameter of 1.4 nm, lengths of 14, 28 and 42 nm	Bridged	Nonlocal EBT	>10^−21^ g
Shen et al. [137]	SWCNTs with diameter of 1.1 nm, lengths of 11, 22, and 33	Bridged	Nonlocal TBT	>10^−21^ g
Natsuki et al. [138]	SWCNTs with diameter of 1 nm, lengths of 10, 20 and 50 nm	Bridged under axial tensile load	EBT	>10^−22^ g
Natsuki et al. [139]	SWCNTs with diameter of 1 nm, length of 20 nm	Bridged under axial tensile load	Nonlocal EBT	>10^−22^ g
Bouchaala et al. [140]	CNTs with diameter of 5 nm, length of 1000 nm	Cantilevered under direct current load	EBT	>7.735 × 10^−21^ g
Eltaher and Agwa [141]	armchair (5,5), (7,7), (10,10), (15,15) and zigzag (5,0), (7,0), (10,0), (15,0) SWCNTs with length of 1.6 nm	Bridged under axial tensile load	MSM, TBT	>10^−22^ g
Eltaher et al. [142]	CNTs with diameter of 5 nm, lengths of 50, 100 and 250 nm	Bridged	Nonlocal EBT	>5 × 10^−21^ g
Ghaffari et al. [143]	CNTs with diameter of 0.8 nm~ 8 nm, lengths of 25, 50, 75 and 100 nm	Bridged under thermal load	Nonlocal EBT	>6.65 × 10^−24^ g
Ghaffari et al. [144]	CNTs with dimensionless parameters	Bridged under thermal load	Nonlocal EBT	>0.218 × 10^−24^ g

**Table 2 sensors-21-01907-t002:** Summarization of studies on graphene sheets-based nano-mass sensors.

Author (s)	Information of GSs	Boundary Condition (s)	Method (s)	Sensitivity
Tsiamaki et al. [146]	Circular SLGSs with diameter of 1 nm~10 nm	Clamped	MSM, FEM	>10^−22^ g
Xu et al. [147]	Rectangular SLGSs of 10 × 5~20 nm	Cantilevered	EPT, FEM	>10^−22^ g
Xu et al. [148]	Rectangular SLGSs of 10 × 5~20 nm	Three cases	EPT, FEM	>10^−22^ g
Natsuki et al. [149]	Rectangular SLGSs of 13.6 × 13.6 nm Rectangular DLGSs of 13.6 × 6.8~27.2 nm	Simply supported	EPT	>10^−22^ g DLGSs is higher than SLGSs
Lei et al. [150]	Circular SLGSs with diameter of 3.4~17 nm	Clamped	EPT	>10^−24^ g
Shen et al. [151]	Rectangular SLGSs of 10~30 nm × 10~30 nm	Simply supported	Nonlocal EPT	>10^−21^ g
Lee et al. [152]	Rectangular SLGSs of 10 × 10 nm	Simply supported	Nonlocal EPT	>10^−27^ g/Hz
Jalali et al. [153]	Rectangular SLGSs with dimensionless parameters	Clamped, simply supported	Nonlocal EBT	Not mentioned
Zhou et al. [154]	Circular SLGSs with diameter of 10 nm, 15 nm, and 20 nm	Clamped, simply supported	Nonlocal EPT	>10^− 21^ g
Natsuki [155]	Rectangular SLGSs of 5.08 × 5.08 nm Rectangular DLGSs of 5.08 × 2.54~10.16 nm	Simply supported	Nonlocal EPT	>10^−22^ g DLGSs is higher than SLGSs
Natsuki et al. [156]	Rectangular SLGSs of 5.08 × 5.08 nm Rectangular DLGSs of 5.08 × 5.08 nm	Simply supported under thermal load	Nonlocal EPT	>10^−22^ g DLGSs is higher than SLGSs
Shen et al. [157]	Rectangular DLGSs of 10 × 10 nm	Clamped, simply supported	Nonlocal EPT	>10^−24^ g
Rajabi and Hosseini-Hashemi [158]	SLGNR of 16 × 2 nm	Cantilevered	EBT	>10^−15^ g
Li et al. [159]	Buckled GNR of 50 × 5 nm	Clamped	Nonlocal EBT, FEM	Not mentioned

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
