# Peer review of "Review on Carbon Nanomaterials-Based Nano-Mass and Nano-Force Sensors by Theoretical Analysis of Vibration Behavior"

_sensors, 2021, doi:10.3390/s21051907_

Round 1
Reviewer 1 Report
The authors reviewed recent studies of frequency-based carbon nano-mass and nano-force sensors using carbon nanomaterials as resonators by continuum mechanical approaches. The review paper is of interest to the readers of Sensors.
The abstract needs to rewritten in a more formal manner.
The manuscript is well organized, however major English editing is recommended (informal expressions, few typos should be checked).
Figure 5a should be provided in a higher image resolution (as Fig.5b)
The references are appropriate, up-to-date and comprehensive. The conclusions of the review are relevant and thoroughly discussed. Overall, the manuscript is worth being published on Sensors after some minor changes.
Author Response
Comments and Suggestions for Authors
The authors reviewed recent studies of frequency-based carbon nano-mass and nano-force sensors using carbon nanomaterials as resonators by continuum mechanical approaches. The review paper is of interest to the readers of Sensors.
The abstract needs to rewritten in a more formal manner.
Reply: Thank you very much for your comments, we have rewritten the abstract as shown in the revised manuscript.
The manuscript is well organized, however major English editing is recommended (informal expressions, few typos should be checked).
Reply: As your comments, we have checked the manuscript thoroughly, details of the revisions are shown in the revised manuscript.
Figure 5a should be provided in a higher image resolution (as Fig.5b)
Reply: Sorry for the low image resolution of Figure 5a. Because Figure 5a is referred from a previous publication, we should adopt it as the present form.
The references are appropriate, up-to-date and comprehensive. The conclusions of the review are relevant and thoroughly discussed. Overall, the manuscript is worth being published on Sensors after some minor changes.
Reply: Thank you very much for your recommendation.
Reviewer 2 Report
Manuscript ID: sensors-1137788
Title: Review on carbon nanomaterials-based nano-mass and nano-force sensors by theoretical analysis of vibration behavior
The manuscript is has been well organized and comprehensively described on the concept of CNTs in nano-mass and nano-force.
Following are the comments:
- I would like the authors to comment on the relevance of the developed theoretical models on graphene sheets and carbon nanotubes towards its specific real-time applications in the conclusion section.
- The authors must add the reference for the images of carbon nanomaterials in Figure 1 if those were not drawn by them.
Author Response
Comments and Suggestions for Authors
The manuscript has been well organized and comprehensively described on the concept of CNTs in nano-mass and nano-force.
Following are the comments:
1. I would like the authors to comment on the relevance of the developed theoretical models on graphene sheets and carbon nanotubes towards its specific real-time applications in the conclusion section.
Reply: Thank you very much for your comments. As your comment, we have added “V. At present, the nanobalance technique for measuring frequency shift of CNTs was demonstrated that could be applied to measure the mass of a tiny particle as light as 22 ×10−15 g [38]. 51 gold atoms loaded on CNTs resonators could be experimentally measured using the relationship between the resonance frequency and atom numbers [40]. However, the real-time application of the nano-testing techniques would be a big challenge due to the small size and weight of carbon nanomaterials. New methods and approaches should be well established to reduce the measurement uncertainly and increase testing accuracy.” (page 15, line 480-486) in the conclusion section of the revised manuscript.
2. The authors must add the reference for the images of carbon nanomaterials in Figure 1 if those were not drawn by them.
Reply: Thank you for your advice, we confirm that Figure 1 was drawn by ourselves.
Reviewer 3 Report
This paper presents the progress of research on carbon nanomaterials in nanoforce and nanomass sensors. The authors first present the mechanism of these two frequency-based nanosensors using mathematical expressions, and then, they summarize the modeling of carbon nanomaterials-based nanoforce and nanomass sensors. They also list some relevant representative works and point out the challenging work ahead. I think this review is written in great detail and can be accepted for publication after minor revision.
Specific comments:
- Section Mechanism of Frequency-Based Nano-Mass and Nano-Force Sensors: This section also needs to be supported by the corresponding reference.
- Did the authors draw Figure 2 themselves? If this figure was borrowed from Ref[100], it needs to be cited.
- Why is the background color of Figure 5a different from the other figures?
- In section2, although the authors list Table 2, I would suggest describing several important works in the body of the text.
- No graphene or carbyne-based nano-force sensor?
- The author needs to correct the heading of Section 5.
- Reference section: The authors included DOIs for only a limited number of articles.
Author Response
Comments and Suggestions for Authors
This paper presents the progress of research on carbon nanomaterials in nanoforce and nanomass sensors. The authors first present the mechanism of these two frequency-based nanosensors using mathematical expressions, and then, they summarize the modeling of carbon nanomaterials-based nanoforce and nanomass sensors. They also list some relevant representative works and point out the challenging work ahead. I think this review is written in great detail and can be accepted for publication after minor revision.
Specific comments:
1. Section Mechanism of Frequency-Based Nano-Mass and Nano-Force Sensors: This section also needs to be supported by the corresponding reference.
Reply: Thank you very much for your comments. We have added references [41,44,69,71] in the first paragraph of Section 2 (page 4, line 146) for supporting this section as your comments.
2. Did the authors draw Figure 2 themselves? If this figure was borrowed from Ref [100], it needs to be cited.
Reply: Thank you for your advice, we confirm that Figure 2 was drawn by ourselves, which is different from the figure shown in Ref [100].
3. Why is the background color of Figure 5a different from the other figures?
Reply: We are sorry about that Figure 5a is referred from a previous publication, so we should adopt its original form, whose background color is different from the other figures.
4. In section2, although the authors list Table 2, I would suggest describing several important works in the body of the text.
Reply: As your suggestion, we have added “Tsiamaki et al. [146] proposed a circular GSs-based nano-mass sensor and simulated its vibration behavior using FEM for calculating the frequency shift. They discussed different boundary conditions of the GSs resonators and compared their results with other works to demonstrate the reasonable accuracy of the results. Their results showed that the proposed nano-mass sensor had sensitivity of 10-22 g level. Natsuki et al. [149] presented a frequency-based nano-mass sensor using rectangular DLGSs as resonators, where a continuum EPT was adopted for vibration analysis and sensitivity of the presented nano-mass sensor could also reach 10-22 g level.” (page 11, line 341-348) and “Shen et al. [151] modeled a simply supported SLGSs-based nano-mass sensor and calculated its frequency shifts using the nonlocal Kirchhoff plate theory. The mass sensitivity of the SLGSs-based nano-mass sensor could reach at least 10-21 g. They also pointed out that the frequency shifts became smaller when the nonlocal effect was considered.” (page 11, line 350-354) in section 4.2 for describing several important works listed in Table 2.
5. No graphene or carbyne-based nano-force sensor?
Reply: Because frequency-based nano-force sensors usually use beam-like resonators, carbon nanotubes are the best candidate, and there has no works on graphene or carbyne-based nano-force sensors to the best of our knowledge. However, as we appealed in the conclusions, graphene sheets or carbyne-based nano-force sensors also deserve to be investigated.
6. The author needs to correct the heading of Section 5.
Reply: Sorry for the mistake, we have corrected the heading of Section 5 as “Nano-Force Sensor”.
7. Reference section: The authors included DOIs for only a limited number of articles.
Reply: As your comment. we have given DOIs of all the references except for Ref. [158] because we could not find its DOI anywhere.